# Plasticity-Enhancing Effects of Levodopa Treatment after Stroke

**DOI:** 10.3390/ijms221910226

**Published:** 2021-09-23

**Authors:** Daniela Talhada, Niklas Marklund, Tadeusz Wieloch, Enida Kuric, Karsten Ruscher

**Affiliations:** 1Laboratory for Experimental Brain Research, Division of Neurosurgery, Department of Clinical Sciences, Lund University, S-22184 Lund, Sweden; daniela.talhada@med.lu.se (D.T.); tadeusz.wieloch@med.lu.se (T.W.); enidakuric@gmail.com (E.K.); 2LUBIN Lab—Lunds Laboratorium för Neurokirurgisk Hjärnskadeforskning, Division of Neurosurgery, Department of Clinical Sciences, Lund University, S-22184 Lund, Sweden; niklas.marklund@med.lu.se

**Keywords:** stroke recovery, dopamine, neuronal plasticiticy, Nogo A, Nogo receptor, oligodendrocyte

## Abstract

Dopaminergic treatment in combination with rehabilitative training enhances long-term recovery after stroke. However, the underlying mechanisms on structural plasticity are unknown. Here, we show an increased dopaminergic innervation of the ischemic territory during the first week after stroke induced in Wistar rats subjected to transient occlusion of the middle cerebral artery (tMCAO) for 120 min. This response was also found in rats subjected to permanent focal ischemia induced by photothrombosis (PT) and mice subjected to PT or tMCAO. Dopaminergic branches were detected in the infarct core of mice and rats in both stroke models. In addition, the Nogo A pathway was significantly downregulated in rats treated with levodopa (LD) compared to vehicle-treated animals subjected to tMCAO. Specifically, the number of Nogo A positive oligodendrocytes as well as the levels of Nogo A and the Nogo A receptor were significantly downregulated in the peri-infarct area of LD-treated animals, while the number of Oligodendrocyte transcription factor 2 positive cells increased in this region after treatment. In addition, we observed lower protein levels of Growth Associated Protein 43 in the peri-infarct area compared to sham-operated animals without treatment effect. The results provide the first evidence of the plasticity-promoting actions of dopaminergic treatment following stroke.

## 1. Introduction

After stroke, the brain has the capacity to remodel neuronal circuits, a process initiated within the first days and weeks after the insult. Albeit to a limited extent, neuronal circuit remodeling contributes to the compensation and recovery of lost neurological function [1]. Previous studies have demonstrated that molecular events underlying spontaneous recovery are also related to structural changes in axons, dendrites and synapses and an increased number and activation of endogenous neural stem cells [2]. Moreover, changes in the extracellular matrix [3], glial cell activation [4] as well as angiogenesis [5] are involved in mechanisms of plasticity throughout the postischemic brain. Although these inherent processes are present, they are foremost slow and insufficient and need to be stimulated in the mature brain.

On the other hand, molecular and cellular mechanisms including the Nogo A pathway suppressing recovery are activated in the postischemic brain [6,7]. Nogo A, a member of the reticulon family, is a molecule that inhibits neuronal outgrowth and dendritic spine formation [8]. While also detected in neurons, it is predominately found on myelin sheets and the cytoplasm of oligodendrocytes. The receptor for the 66-residue portion of Nogo A (NgR1) is specifically found on neurons [9]. NgR1 binds a special amino acid sequence of the oligodendrocyte-derived Nogo A called Nogo-66 accumulating on growth cones of central nervous system (CNS) neurons. Contact between growth cones with oligodendrocyte-derived Nogo-66 causes the collapse of neuronal growth cones and thereby failure to form neuronal connections, ultimately inhibiting brain plasticity and recovery after stroke. Targeted treatment with an antibody that inhibits the inhibitory effects of Nogo A increases the sprouting of axons and the regeneration of lesioned descending spinal axons, which increases locomotor recovery after stroke without changes in Nogo A expression [7,10,11,12].

Treatment with the dopamine precursor molecule levodopa (LD) improves recovery after experimental stroke [13,14], attributed to various processes including neuronal plasticity [15], increased production of glia-derived growth factors [16] and the attenuation of inflammatory cascades [17,18] in the ischemic territory. Moreover, adjuvant treatment with LD together with enhanced physical therapy protocols have shown efficacy to enhance functional outcome in stroke patients [19]. In contrast, dopaminergic treatment administered only with standard rehabilitation protocols exerts no beneficial effects [20].

An increased expression of dopamine receptors on different neuron populations and glial cells [13,14,21] prompted us to study the dopaminergic innervation in the ischemic territory. Here, we found an increase of tyrosine hydroxylase (TH) positive fibers in the peri-infarct area in rats and mice subjected either to transient or permanent focal ischemia. Remarkably, individual TH positive branches were detected in the ischemic core tissue. Treatment with LD reduced the number of Nogo A positive mature oligodendrocytes in the peri-infarct area and increased the number of oligodendrocyte transcription factor (OLIG-2) positive cells in this region. Together, we show for the first time the re-innervation of necrotic scar tissue following ischemic stroke, an increased dopaminergic innervation in the ischemic territory associated with a downregulation of the Nogo A pathway potentially beneficial to reorganizing neuronal networks in the ischemic territory.

## 2. Results

### 2.1. Increased Dopaminergic Innervation in the Ischemic Territory Following Experimental Stroke

The expression of dopamine 1 and dopamine 2 receptors in the peri-infarct area of mice subjected to stroke [14] prompted us to investigate the dopaminergic innervation in this region in different animal models of experimental stroke. A fine and filigree network of TH+ fibers was observed in the proximal peri-infarct region in rats subjected to transient Middle Cerebral Artery Occlusion (tMCAO) (Figure 1B) in contrast to sham animals (Figure 1C). TH positive fibers in the ipsilateral Substantia nigra of the same animal are shown as the positive control (Figure 1D). In addition to the fibers in the peri-infarct area, higher magnification micrographs revealed thin (thickness < 1 µm) TH positive branches located in the ischemic infarct core as early as 5 days following tMCAO (Figure 1E,E1). We analyzed the levels of TH in the ischemic brain in rats subjected to sham operation with rats subjected to tMCAO (Figure 1F) and found higher levels of TH as we observed higher levels in the lesioned hemisphere compared to homotypic regions of the contralateral hemisphere (Figure 1G). Treatment with LD did not affect TH levels in the peri-infarct area (Figure 1H). An increased dopaminergic input was observed in rats and mice subjected to permanent focal ischemic induced by photothrombosis (PT) (Figure 2A,B) as well as mice subjected to tMCAO during the first 14 days after stroke onset (Figure 2C). Together, this set of data shows an increased dopaminergic innervation potentially relevant for multiple mechanisms, e.g., plasticity, gliosis and inflammation in the ischemic territory.

### 2.2. Treatment with Levodopa Reduces the Number of Nogo A+ Mature Oligodendrocytes in the Peri-Infarct Area

Increased levels of growth inhibitory proteins and the activation of respective cascades in the peri-infarct zone essentially block neuronal outgrowth [23] and synaptogenesis [24]. As shown previously by others, we observed an increase of Nogo A positive mature glutathione-S-transferase π (GST π) positive oligodendrocytes in the peri-infarct area following tMCAO (Figure 3A) compared to rats subjected to sham surgery (Appendix A). A significant lower number of Nogo A+/GST π+ oligodendrocytes in the peri-infarct area were found in rats treated with LD 20 mg/kg for 12 days after tMCAO (Figure 2B). Correspondent to the reduced number of cells, lower levels of Nogo A were found in this region (Figure 2C). In sham-operated rats treated either with saline or LD, no difference was observed in the quantification of Nogo A+ oligodendrocytes (Appendix A Appendix A).

The increased number of TH positive neuronal branches also suggests a reduced expression of molecular inhibitors on neurons, i.e., the NgR1. Following tMCAO, treatment with LD (20 mg/kg) significantly reduced the level of NgR1 in the peri-infarct area (2.17 ± 0.58; *n* = 8) compared to the saline-treated vehicle group (AU 3.91 ± 0.39; *n* = 8) (Figure 4A).

The appearance of dopaminergic fibers in the peri-infarct area further prompted us to study the level of Growth-Associated Protein 43 (GAP 43) in rats subjected to tMCAO and treated either with LD or saline and sham-operated rats, respectively. The analysis of the samples from the peri-infarct area of injured rats treated either with vehicle or LD showed lower levels of GAP 43 (AU tMCAO vh 1.66 ± 0.52, *n* = 6; tMCAO LD 0.60 ± 0.16, *n* = 6) compared to the sham-operated animals (AU 3.13 ± 0.54, *n* = 5) (Figure 4B). These results for the first time provide experimental evidence that the Nogo A pathway can be targeted pharmacologically with LD, a pharmacological treatment.

### 2.3. Increased Number of Adult Oligodendrocyte Precursor Cells in the Peri-Infarct Area of Rats Treated with Levodopa Following tMCAO

The role of the dopaminergic system on oligodendrogenesis is unknown, in particular following an acquired brain injury such as ischemic stroke. Substantial tissue loss due to primary and secondary cell death results in the disruption of neuronal connections and circuitries and is accompanied by myelinating cells. Structural changes of neurons in healthy tissue in the lesioned hemisphere are well documented [25], hence, it is not exactly known if and how myelinating cells regenerate or are replaced. In rats submitted to tMCAO, we found OLIG-2 positive oligodendrocyte precursor cells around the infarct core 14 days after the insult (Figure 5A,B). Cells mainly accumulated in layers 2 and 3 and layers 5 and 6 (Figure 5C,D) with higher magnifications in Figure 5C1,D1 showing a glia-like stellate morphology. The quantification of cells revealed a significant higher number of OLIG-2 positive cells in rats treated with LD (median 261 cells, Q1 245 cells, Q3 307 cells) compared to saline-treated vehicle animals (median 132 cells, Q1 116 cells, Q3 138 cells) (Figure 5B). These numbers show a clear effect in the levodopa-treated rats.

## 3. Discussion

The present study has been conducted to further understand the role of dopamine signaling in the mechanisms of plasticity in the postischemic brain. To the best of our knowledge, this is the first ever study reporting an increased dopaminergic innervation of the ischemic territory during the first two weeks after stroke. In addition, we found that treatment with LD significantly downregulates the Nogo A pathway accompanied by lower GAP43 protein levels in rats following tMCAO. Our study also supports the idea that dopaminergic pathways modulate oligodendrogenesis in regions of increased loss of myelinating cells such as the proximal peri-infarct area.

### 3.1. Increased Dopaminergic Innervation in the Ischemic Territory

We found an increased number of TH positive neuronal fibers in the ischemic territory both following transient and permanent focal ischemia, both in rats and mice similarly. This finding across species and stroke models suggests an inherent process to increase the dopaminergic innervation of the ischemic territory during the first weeks after stroke. More remarkably, we found individual dopaminergic fibers present in the ischemic infarct core during the first seven days after the insult. The low number of fibers indicates that a permissive cellular and molecular environment within the critical time window is required for neuronal branching into the necrotic tissue of the infarct core. During the first days after the insult, essentially, an immature scar containing microglia and reactive astroglial cells had formed [26]. In addition, surviving cells inside the infarct core would need to provide molecular cues for guidance or even neurotrophic support. Indeed, we have previously identified a population of nestin positive astrocytes expressing glial-derived neurotrophic factor (GDNF) in the infarct core [16]. Overall, the branching of dopaminergic fibers might be regulated by increased levels of neurotrophic factors including GDNF [16] and brain-derived neurotrophic factor (BDNF) [27,28] released by glial cells.

Interestingly, dopaminergic innervation is observed with a higher density of microvessels towards the infarct core during the first weeks after stroke [5,29]. Accompanying angiogenesis, mediators are released from endothelial cells that promote trophic effects and neuronal sprouting such as vascular endothelial growth factor (VEGF) [30] and laminin/β1 integrins [5]. Indeed, in addition to its pro-angiogenic effects, integrin α5β1 enhances neurite the outgrowth of striatal dopaminergic neurons [31]. An increased dopaminergic innervation in the peri-infarct area may stimulate relevant processes of tissue reorganization. As we have previously shown, dopaminergic cascades are involved in the modulation of poststroke astrogliosis [13], the attenuation of poststroke inflammatory cascades [18] and immunodepression [32]. Moreover, dopaminergic signaling promotes angiogenesis as shown in preclinical models of Parkinson’s disease [33] and might be essential for functional improvement through VEGF-mediated angiogenesis after stroke [12].

### 3.2. Effects of Levodopa Treatment on Nogo A Pathway and GAP 43

Dopamine actions are dependent on the spatiotemporal profile of receptor surface expression on various cell types. Differentiated oligodendrocytes express dopamine 2 and 3 receptors [21]. Mature oligodendrocytes are one of the main sources of Nogo A, a myelin-associated protein and neurite growth inhibitor of the reticulon family [6,7]. We found that treatment with LD significantly affects the Nogo A pathway with potential relevance for neuronal plasticity processes in this area and the entire injured brain. Levodopa treatment downregulated the number of Nogo A positive mature oligodendrocytes, accompanied by decreased levels of Nogo A protein levels in comparison to animals subjected to tMCAO and treated with saline. Following treatment, reduced levels of Nogo A, indirectly or directly, may have positive effects on the regulation of neurite outgrowth and sprouting [34], dendritic branching and dendritic spine formation and elimination [35] and may also experience dependent plasticity after the insult [36].

In addition, we found lower levels of the Nogo A receptor in the peri-infarct area of rats treated with LD after tMCAO either due to degradation or reduced de novo synthesis. Therefore, the function of other myelin-associated inhibitors such as myelin-associated glycoprotein (MAG) and oligodendrocyte myelin glycoprotein (OMgp) binding to the NgR1, with similar effects as described for Nogo A [37], might have been abrogated. Mechanistically, the increased levels of cyclic adenosine monophosphate (cAMP) result in the internalization of NgR1 [38]. Compounds that increase intracellular cAMP, i.e., the phosphodiesterase 4 inhibitor rolipram or the adenylate cyclase activator forskolin, enhance this process independent from the protein kinase A. Although an increase in cAMP has been described as fast and reversible, a prolonged increased dopaminergic tonus may result in the degradation of NgR1 mediated through dopamine 1 receptors. This seems plausible since lower levels of NgR1 were found in whole cellular extracts, and no differences would have been obtained if NgR1 would have been internalized into endosomes [38]. Further experiments will be required to elucidate the exact dynamics of NgR1 upon dopaminergic treatment over longer intervals and including investigations to study the balance between activation of dopamine 1/5 receptors, which increase the level of cAMP, and the activation of dopamine 2/3 receptors, which result in reduced cAMP levels.

Together, the inhibition of the Nogo A pathway by desensitizing neurons to axonal growth inhibitors by downregulation of the NgR1 as well as scavenging Nogo A have beneficial effects on the recovery of lost function after experimental stroke [12]. Here, we show that rats subjected to tMCAO and treated with LD showed a significant downregulation of NgR1 and Nogo A; these cohorts showed a significant recovery without affecting the lesion size as we have reported previously [13,22]. Therefore, enhancing dopaminergic signaling following stroke may be a promising adjuvant to treatments aiming at reducing the levels of myelin-associated inhibitors.

On the other hand, signaling pathways have been identified that stimulate and regulate neuronal outgrowth. Integrated GAP 43 is involved in the regulation of axonal sprouting, synaptic function and nerve regeneration [39]. An increased expression of GAP 43 was found in the peri-infarct area during the first weeks after the insult [40,41]. An additional upregulation in the ipsilateral somatosensory cortex of mice subjected to tMCAO was observed after optogenetic stimulation of the dentato-thalamo-cortical pathway [42]. Here, we found that rats subjected to tMCAO had lower levels of GAP 43 compared to sham-operated animals. No difference was found between the treatment groups after stroke. This indicates that the expression of GAP 43 in our model is dopamine independent. Opposite results compared to previous studies might be related to different expression kinetics or feedback regulation of protein expression and analysis of GAP 43 on the transcript and protein level.

### 3.3. Levodopa Treatment and Oligodendrogenesis

Our experiments showed a non-significant decrease in the number of oligodendrocytes in the peri-infarct area after treatment with LD, compared to vehicle treatment after tMCAO reasoning reduced levels and number of Nogo A positive cells in this region. Further evaluation of oligodendrocytes in the peri-infarct area revealed an increase in the number of OLIG-2 positive cells. From these results, we conclude that dopamine signaling may contribute to oligodendrocyte proliferation and maturation. The turnover of mature oligodendrocytes in the brain areas around the lesion seems reasonable due to structural reorganization of neuronal circuits and connections. The underlying mechanisms are unknown and will be unraveled in future studies and may include investigations on direct catecholamine toxicity [43] or indirect cytotoxicity mediated by increased neuronal activity and through ionotropic glutamate receptors [44].

The trophic effect of dopamine signaling is supported by studies showing that anti-psychotic treatments targeting the dopaminergic system (i.e., olanzapine) resulted in the upregulation of a number of genes linked to myelination and oligodendrocyte development [45]. In addition, the Wnt/β-catenin beta pathway is related to oligodendrogenesis and reduced phosphorylation of β-catenin via the protein kinase B/glycogen synthase kinase 3 beta pathway may contribute to increased differentiation of oligodendrocyte precursor cells. Hence, dopamine 2 receptors modulate Wnt expression and thereby control cell proliferation [46]. Interestingly, the number of oligodendrocytes in sham-operated animals was not different between the treatments. This suggests that oligodendrocytes, respective precursor cells, in the peri-infarct area change their susceptibility to dopamine possibly by changing the dopamine receptor profile following stroke. Together, our data point towards rejuvenating actions of dopamine signaling in oligodendrocytes in the peri-infarct area.

## 4. Materials and Methods

### 4.1. Experimental Design

This study is a follow-up derived from investigations including larger cohorts of rats and mice, from different experimental studies. All the studies described had been approved by the regional ethical committee.

Study I [13] was performed to investigate the effects of dopamine treatment on functional recovery after experimental stroke. In brief, 26 Wistar rats were assigned to experimental groups and thereafter subjected to experimental stroke induced by tMCAO or sham operation. On day 2, animals were treated either with Levodopa (LD 20 mg/kg) in combination with benserazide (15 mg/kg) dissolved in saline solution or saline (vehicle) for 12 days by daily i.p. injections. Animal groups included in the study were tMCAO LD *n* = 8, tMCAO vehicle *n* = 8, Sham LD *n* = 5 and Sham vehicle *n* = 5. On day 14, after surgeries, animals were sacrificed for endpoint analyses (Figure 1).

Sixteen adult Wistar rats included in study II [22] were subjected to permanent MCAO (pMCAO, *n* = 8) or Sham (*n* = 8) and sacrificed five days after for endpoint analyses (Figure 1).

To evaluate dopaminergic innervation in the ischemic territory, we have used rats and mice subjected either to PT or tMCAO [47] and one mouse after tMCAO [48].

### 4.2. Transient/Permanent Occlusion of the Middle Cerebral Artery (tMCAO/pMCAO) and Photothrombosis (PT)

Transient MCAO (tMCAO) was induced in male Wistar rats for 120 min, for Study I [13]. In brief, the rats were anesthetized (initial 4% fluothane in N_2_O/O_2_ [70:30]; maintenance 2% fluothane in N_2_O/O_2_ [70:30]). Following preparation of the carotid bifurcation and ligation of the external carotid artery, a nylon filament was inserted into the origin of the MCA through the internal carotid artery. Regional blood flow in the brain was monitored by a laser Doppler probe mounted on the skull above the MCA region (PeriFlux System 5000; Perimed, Jarfalla, Sweden). After the time period of occlusion, the filament was removed. Sham animals were equally operated, excepting the introduction of a filament into the MCA. The same procedure was performed in mice as described before [48,49], with time occlusion period of 45 min.

Permanent MCAO (pMCAO) was induced in male Wistar rats as described previously [22]. Summarizing, rats were anesthetized and the right MCA permanently ligated. In sham-operated animals, the MCA was exposed but not ligated.

PT in rat and mouse was performed as described previously [47,50]. Summarizing, animals were anesthetized (initial 4% isoflurane in N_2_O/O_2_ [70:30]; maintenance 2% isoflurane in N_2_O/O_2_ [70:30]) and the skull exposed. Stroke was induced after intravenous (rat) or intraperitoneal (mouse) injection of the photosensitive dye Rose Bengal (0.5 mL at 10 mg/mL for rat; 0.1 mL at 10 mg/mL for mouse; Sigma-Aldrich, Taufkirchen, Germany) and the right hemisphere exposed with a cold light for 20 min (Schott KL 1500 LCD, intensity: 3050–3200 K).

### 4.3. Western Blots

After euthanasia, brains were immediately collected and fresh frozen in isopentane −70 °C. Proteins from the brains were extracted as described in each experimental study included [13,22].

Ten to twenty micrograms of protein were mixed with lysis buffer to a total volume of 10 µL. Another 10 µL of a 2× sample buffer (125 mM Tris HCl pH 6.8, 4% SDS (sodium dodecyl sulfate), 20% glycerol, 100 mM Dithiothreitol (DTT), 0.2% bromophenol blue were added for a final volume of 20 µL. Proteins were denatured at 95 °C and separated in 10%, 4–15% polyacrylamide gradient gels or precast 4 to 15% Mini-PROTEAN TGX gradient PAGE gels and any kD Mini-PROTEAN TGX Stain-Free gels (Bio-Rad, Solna, Sweden). Following transfer, membranes were blocked with 5% non-fat dry milk in tris buffered saline mixed with 0.1% Tween-20 (TBST) for one hour at room temperature (rt). Thereafter, membranes were incubated with the primary antibody (Table 1) in 5% bovine serum albumin in TBST at 4 °C overnight. The next day, membranes were incubated with respective secondary antibodies (Table 1) for one hour at rt in the blocking solution.

After several washing steps with TBST, signals were visualized by using a chemiluminescent HRP substrate and a LAS1000 camera system (Fuji, Tokyo, Japan) or a Chemidoc MP system (Bio-Rad, Solna, Sweden).

After exposure and washing, membranes were stripped in respective buffer at 70 °C for 20 min and reprobed for β-actin (anti β-actin HRP conjugated antibody, diluted at 1:75,000) assumed to be stable in all treatment conditions. The expression of proteins of interest was calculated as a ratio of β-actin by using ImageJ or Image Lab Software v6.0.1 (Bio-Rad). For GAP 43 Western blots, total protein normalization was performed using a stain-free enabled imager (Bio-Rad, Solna, Sweden).

Representative fluorescent Western blots were incubated with StarbrightBlue 700 fluorescent secondary anti-rabbit (Bio-Rad, cat#12004162) or StarbrightBlue 520 fluorescent secondary anti-mouse (Bio-Rad, cat# 12005866), both diluted at 1:5000, for one hour at rt in the blocking solution. As a loading control, Rhodamine anti-glyceraldehyde 3-phosphate dehydrogenase (GAPDH) antibody (Bio-Rad, cat#12004168) was incubated together with the secondary antibody, diluted at 1:10,000. Signals were visualized by using a Chemidoc MP system (Bio-Rad, Solna, Sweden).

### 4.4. Immunofluorescence/Immunohistochemistry

At the endpoint of the studies, animals used for immunofluorescence or immunohistochemistry analysis were perfused fixed with PFA 4%, as described for each individual study [13,22,47,48]. Thirty-micrometer-thick sections were used from rats and mice from the different studies (see above and respective figure legends). Sections from 1.2 to 1.0 mm anterior to bregma were chosen for stainings.

For immunofluorescence, the sections were washed in phosphate buffered saline (PBS) 3 × 10 min. Thereafter, they were blocked in PBS containing 5% normal donkey serum (NDS) and 0.25% Triton X-100 (TX100) for one hour at rt. Thereafter, sections were incubated with primary antibodies mixed in blocking solution over night at 4 °C. Primary antibodies used are described in Table 1. After incubation with the primary antibody, the sections were rinsed 3 × 10 min with PBST at rt and incubated with the secondary antibodies for one hour at rt protected from light. The sections were washed 3 × 10 min with PBST at rt while protected from light. The sections used for the Nogo A/GST π staining were incubated for one hour at rt, protected from light, with a streptavidin Alexa Fluor 488 solution and then rinsed 3 × 10 min with PBST. Afterwards, the sections were mounted on charged slides and allowed to dry in darkness and then cover slipped. After the sections had dried, the fluorescent signals were visualized using a confocal microscopy system (LSM510, Carl Zeiss, Germany).

For immunohistochemistry, protocol was performed as for immunofluorescence, added the step that sections were quenched before blocking to block endogenous peroxidases in a PBS-solution mixed with 3% hydrogen peroxide and 10% methanol for 20 min at rt. Sections were incubated in primary antibodies (Table 1). The next day, the sections were rinsed 3 × 10 min with PBST and then incubated with the anti-rabbit biotinylated secondary antibody (1:200) in PBST for 1 h at rt on a shaker. After rinsing with PBST 3 × 10 min, the sections were incubated with an ABC-solution (10 µL A and 10 µL B/1 mL PBST from an ABC-HRP kit) mixed with PBST for 1 h at rt. An NiDAB-solution (20 µL Dabsafe with 4.5 µL 8% NiCl_2_/1 mL PBS) was prepared and used to stain the sections after they had been rinsed 3 × 10 min with PBS. The sections were incubated for 1 min in the NiDAB-solution and then 3% H_2_O_2_ in purified water was added to start the staining reaction. After 30 s in this mix, the sections were rinsed 3 × 10 min in PBS. The sections were mounted on charged slides and dehydrated after being air dried and coverslipped with pertex.

To quantify the number of Nogo A and GST π and OLIG-2 positive cells in the cortical peri-infarct, an area of 500 pixels adjacent to the infarct area was defined and cells were counted manually based on cell morphology (Figure 5A).

### 4.5. Statistical Analysis

Data are expressed as means ± standard error of the mean (SEM). A value of *p* < 0.05 was considered as statistically significant. Statistical analysis was performed using GraphPad Prism 6.0 software (GraphPad, San Diego, CA, USA). For the comparison of two groups, unpaired Student’s *t*-test and Mann–Whitney two-tailed test were used for parametric and non-parametric data, respectively. One-way ANOVA with Bonferroni correction was used for more than two groups.

## 5. Conclusions

In the present study, we have demonstrated that dopamine significantly downregulates the level of Nogo A in the postischemic hemisphere of rats subjected to experimental stroke. Moreover, we found that the number of Nogo A positive oligodendrocytes was significantly reduced in LD-treated animals. In addition, treatment significantly enhanced the number of oligodendrocyte precursor cells in the proximal peri-infarct area. These results are promising to further investigate LD as a recovery-enhancing medicine in stroke rehabilitation.

## Figures and Tables

**Figure 1 ijms-22-10226-f001:**
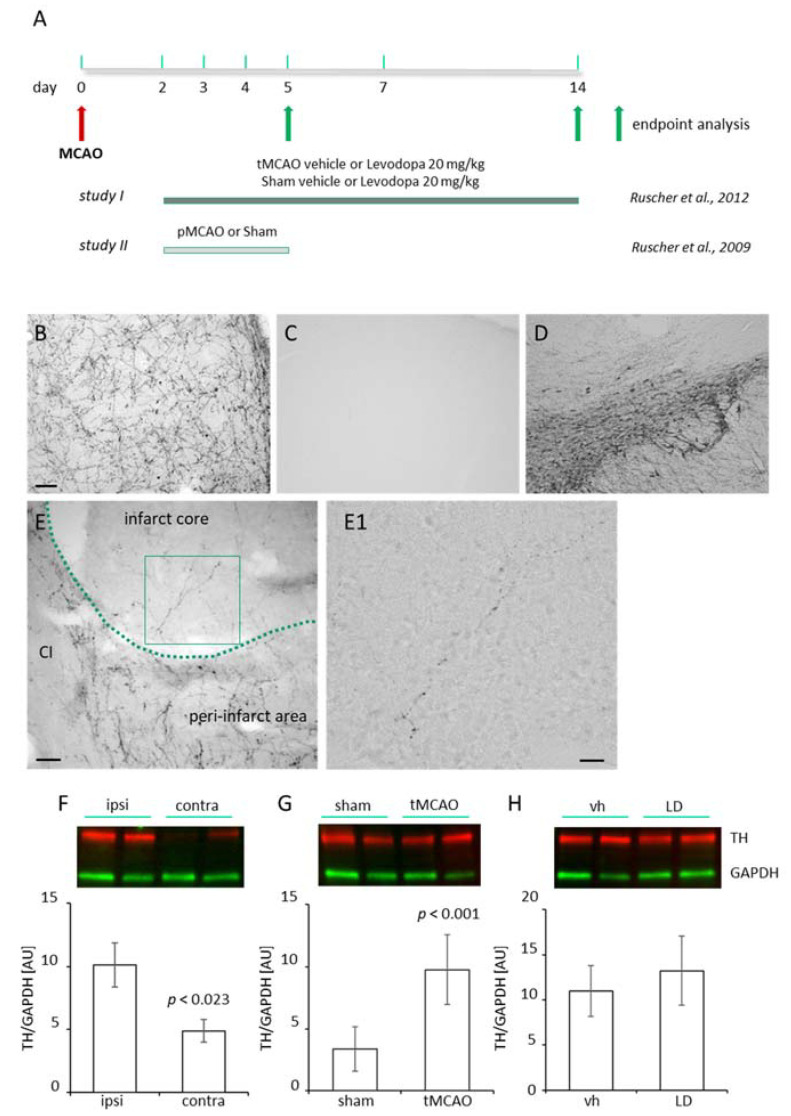
(**A**) Brain tissue sections and brain lysates from regions of interest were obtained from male Wistar rats subjected to transient or permanent occlusion of the middle cerebral artery (tMCAO/pMCAO) [13,22]. At the experimental endpoints (study I: 14 days; study II: 5 days) brains were either analyzed for dopaminergic innervation (study II), the Nogo A pathway, the number oligodendrocyte precursor cells and Growth-Associated Protein 43 (GAP 43) protein levels (study I). (**B**) Dopaminergic innervation in the peri-infarct area five days following pMCAO and (**C**) in the cortex of a sham-operated rat. (**D**) The ipsilateral Substantia nigra of the same animal shown in (**C**) as positive control for tyrosine hydroxylase (TH). Scale bars (**B**–**D**)—100 µm. (**E**) Higher magnification of the proximal peri-infarct/infarct core border zone. The infarct border is delineated as a green dashed line. CI: capsula interna, scale bar: 50 µm. (**E1**) Higher magnification of the insert of an individual TH positive branch shown in (**E**). Scale bar: 20 µm. (**F**) TH levels (red, StarbrightBlue 700) in the peri-infarct area (*n* = 8) compared to the homotypic region of the contralateral cortex (*n* = 8), 14 days after tMCAO in rats (**G**) TH levels in the somatosensory/motor cortex of sham operated rats (*n* = 5) compared with rats subjected to tMCAO (*n* = 6). (**H**) Levels of TH in the peri-infarct area of saline (vehicle; *n* = 8) and levodopa (LD 20 mg/kg; *n* = 8) treated rats, 14 days after tMCAO. Levels of TH have been normalized against glyceraldehyde 3-phosphate dehydrogenase (GAPDH, green, rhodamine). Statistical analysis was performed using Student’s *t*-test. *p* value < 0.05 was considered significant and differences are shown in respective figures.

**Figure 2 ijms-22-10226-f002:**
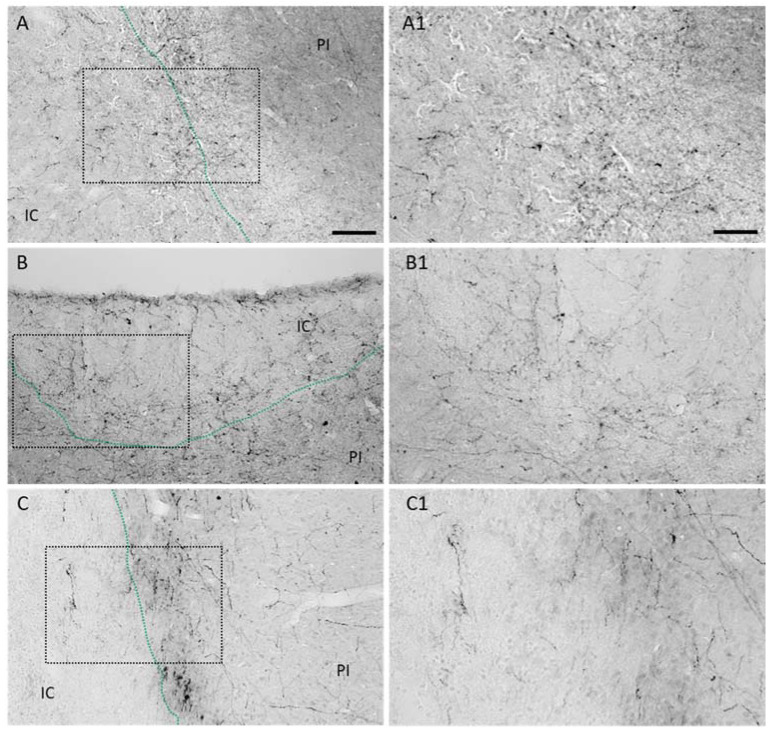
Dopaminergic innervation in rodent models of experimental stroke. Representative micrographs illustrating increased number of tyrosine hydroxylase positive neuronal branches in the ischemic core and peri-infarct of (**A**) rats 21 days after permanent focal stroke induced by photothrombosis (PT), (**B**) 14 days after mice have been subjected to PT or (**C**) to transient MCAO. Higher magnifications of depicted rectangles are shown in (**A1**), (**B1**) and (**C1**), respectively. Scale bars: A, B and C 100 µm, A1, B1 and C1 50 µm.

**Figure 3 ijms-22-10226-f003:**
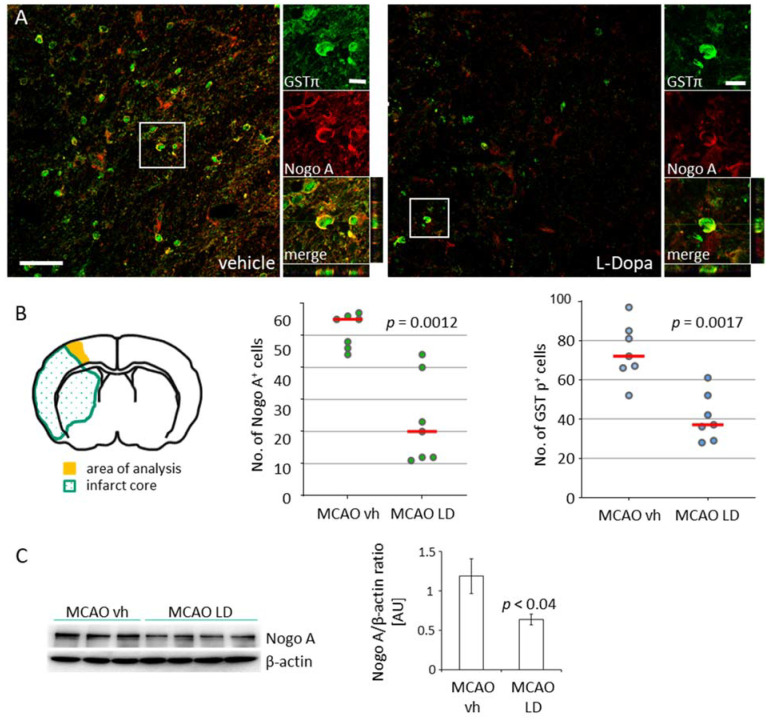
Downregulation of Nogo A by treatment with levodopa. (**A**) Co-staining for Nogo A (red, Cy3) and glutathione-S-transferase π (GST π, green, AF488) in the peri-infarct area of rats treated either with saline (vehicle) or levodopa (LD 20 mg/kg) 14 days after tMCAO; scale bars: low magnification 50 µm, high magnification 10 µm. (**B**) Illustration of the peri-infarct area (yellow) used for analyses and the number of Nogo A and GST π positive cells (*n* = 7, each experimental group) in this region. Statistical analysis was performed by the Mann–Whitney test. (**C**) Levels of Nogo A in the peri-infarct area in rats treated either with vehicle (*n* = 7) or levodopa (*n* = 8) and semi-quantitative analysis of bands after normalization against beta-actin. Statistical analysis was performed by the Student’s *t*-test. *p* Value < 0.05 was considered significant and differences are shown in respective figures.

**Figure 4 ijms-22-10226-f004:**
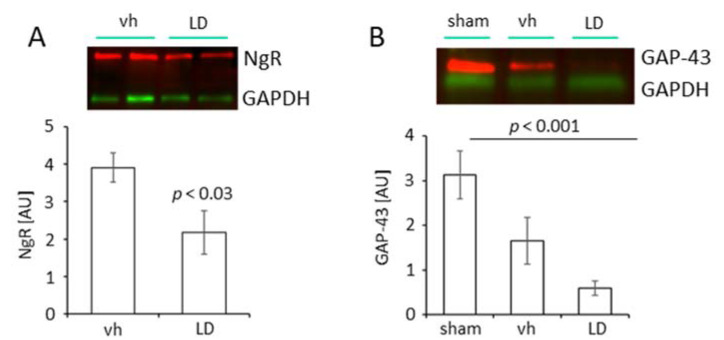
Regulation of Nogo A receptor (NgR) and Growth Associated Protein 43 (GAP 43) following treatment with levodopa. (**A**) Levels of NgR in the proximal peri-infarct area of vehicle (saline; *n* = 8) and levodopa-treated rats (LD 20 mg/kg; *n* = 8) 14 days after tMCAO. Data are presented as mean ± SEM, statistical analysis was performed using Student´s *t*-test. (**B**) Levels of GAP 43 in the proximal peri-infarct area in sham-operated rats treated with saline (*n* = 5) and rats subjected to tMCAO and treated either with saline (*n* = 6) or levodopa (LD 20 mg/kg; *n* = 6). Statistical analysis was performed by One-way ANOVA with posthoc Bonferroni correction. *p* Value < 0.05 was considered significant and differences are shown in respective figures.

**Figure 5 ijms-22-10226-f005:**
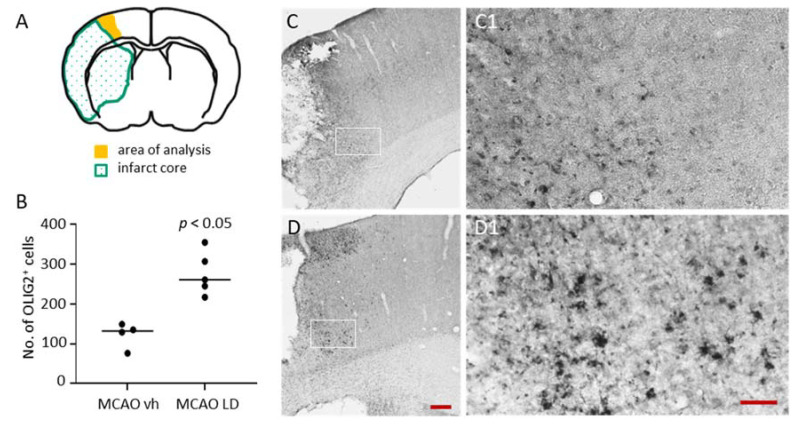
Impact of levodopa treatment on OLIG-2 positive cells following tMCAO. (**A**) Quantification of Oligodendrocyte transcription factor (OLIG-2) positive cells in the proximal peri-infarct area (**B**) with quantification of cells in vehicle (saline; *n* = 4) and levodopa-treated rats (LD 20 mg/kg; *n* = 5) 14 days after tMCAO. Statistical analysis was performed by the Mann–Whitney test. (**C**) Illustration of OLIG-2 positive cells in the peri-infarct area of a rat treated with vehicle (saline) and (**D**) in a rat treated with levodopa (20 mg/kg) 14 days after the insults. Higher magnification of the inserts in (**C**) and (**D**) are shown in (**C1**) and (**D1**), respectively. Scale bars: C and D—200 µm, C1 and D1—50 µm. Regulation of Nogo A receptor (NgR) and Growth-Associated Protein 43 (GAP43) following treatment with levodopa.

**Table 1 ijms-22-10226-t001:** List of primary and secondary antibodies used for Western blot (WB), Immunofluorescence (IF) and Immunohistochemistry (IHC).

Primary Antibody	Supplier/Cat#	Application/Dilution	Secondary Antibody/Dilution
Anti-Nogo A	R&D Systems, cat# AF3098	IF, 1:100	Anti-goat Cy3, 1:400
		WB, 1:500	Anti-goat biotinylated, 1:4000 ; Biotin-HRP 1:3000
Anti-Nogo Receptor	Millipore, cat# AB15138	WB, 1:2000	Anti-rabbit HRP, 1:15,000
glutathione-S-transferase π	BD Transduction Laboratories, cat# 610719	IF, 1:200	Anti-mouse biotinylated 1:400; streptavidin AF488, 1:400
(GST π)			
Growth Associated Protein	Millipore, cat# MAB347	WB, 1:1500	Anti-mouse HRP, 1:10,000
(GAP 43)			
OLIG-2	R&D Systems, cat# AF2418	IHC, 1:200	Anti-rabbit biotinylated, 1:400
Tyrosine hydroxylase (TH)	Millipore, cat# AB152	IHC, 1:1000	Anti-rabbit biotinylated, 1:400
		WB, 1:5000	Anti-rabbit HRP, 1:15,000

## Data Availability

Data is contained within this article and Appendix A.

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
