# Peer review of "Plasticity-Enhancing Effects of Levodopa Treatment after Stroke"

_ijms, 2021, doi:10.3390/ijms221910226_

Round 1

Reviewer 1 Report

SUMMARY

                The submission from Talhada et al. builds on previously published articles from the corresponding author (Refs. 4, 13, 14, 16-18, 31, 38, 47-50). Previous studies have shown levodopa contributes to recovery of sensory motor function in post-stroke recovery, a decrease in neuroinflammation, and increased production of glial derived neurotrophic factor.  This study assesses dopaminergic innervation in rats and mice post-stroke induced by middle cerebral artery occlusion and photothrombosis. The authors show dopaminergic innervation in the infarct core and peri-infarct region within the first two weeks, and treatment with levadopa/benserazide resulted in downregulation of the NOGO-A pathway, increased oligodendrogenesis, and some re-innervation of the necrotic scar tissue. Overall, the manuscript is well-written, scientifically sound, and provides an important contribution to our understanding of the molecular mechanism of levodopa in post-stroke recovery.

Minor points

  1. The article has a few minor grammatical errors that should be cleaned up prior to publication (for example, line 257 has an additional A after growth inhibitors).
  2. The authors should provide the abbreviation after first use, rather than later (see GDNF line 209 vs. 211).

Author Response

Ms #ijms-1335292

Response to reviewer #1

Point 1: “The article has a few minor grammatical errors that should be cleaned up prior to publication (for example, line 257 has an additional A after growth inhibitors).”

We agree, the manuscript has been edited by native English speaker.

Point 2: “The authors should provide the abbreviation after first use, rather than later (see GDNF line 209 vs. 211).”

The manuscript has been changed accordingly.

Reviewer 2 Report

In the present study the authors show that levodopa (LD) significantly downregulates Nogo-A in the ischemic hemisphere in experimental stroke. Likewise, the number of Nogo-A positive oligodendrocytes (and Nogo-A receptors) was significantly reduced in LD-treated animals compared to vehicle, and oligodendrocyte precursor cells in the proximal peri-infarct area were increased.  Parallel to other studies, an increase in dopaminergic innervation of the ischemic territory was observed during the first week of stroke in both rats and mice and this effect is suggestive of the reduced inhibition to growth cone extension. The lower level of GAP-43 in vehicle or LD treated rats as compared to sham is however quite perplexing as this contradicts the increased branching and plasticity as observed in the increased tyrosine hydroxylase positive neuronal branches after stroke.

The authors highlight the relevance of LD treatment during recovery from stroke through thus far unknown mechanisms that promote plasticity that could be very important in the field of recovery and neurorehabilitation as a result of its functional implications.  

Main Concerns

  • Olig2 shows phenotype overlap in the various developmental stages of the oligodendrocyte lineage and is featured from the oligodendrocyte progenoitor cell right up to the mature myelinating oligodendrocyte. Therefore, Olig2 is not entirely specific to adult oligodendrocyte precursors. The bipolar phenotype and more specific PDGF and A2b5 are more reliable markers that should have been used. Therefore it might be incorrect to state that in this study, LD treatment enhanced the number of oligodendrocyte precursor cells after MCAO. It has been extensively shown from previous studies that pre-ols increase in number after MCAO and to the extent of even populating and spreading into the contralateral hemisphere depending on the extent of the severity of the insult. In that case, the authors must retract their notion that DA signaling is directly involved in oligodendrogenesis.

  • The surprising fact that levels of GAP43 was found to be lower after tMCAO compared to sham and subsequently in the treatment group requires a more insightly perspective as this goes against the notion of sprouting and plasticity that are central in regeneration. It is is well established that GAP-43 levels of expression increase during regeneration in growth cones of axonal membranes. As a matter of fact, deletion of one allele of the GAP-43 gene in humans leads to cognitive impairment and failure to form telencephalic commissures.

Minor concerns

  • It is unclear why the authors of this paper performed experiments on two different rodent stroke models that involved both mice and rats. Was it to show congruent results irrespective of animal spieces or was it to determine whether mechanistically their findings can be validated in stroke in general ? In my opinion this point has to be clarified for the sake of the readers of this journal.

  • Was the blood flow reduction in the MCA region and the lesion volumes comparable in all the experimental groups tested ? The authors should at least show some TTC-stained serial sections.
  • Section 2.3 should read: ‘Treatment with levodopa increases the number of adult oligodendrocyte precursor cells in the per-infarct area.
  • LD was selected at a doseage of 20mg/Kg. What was the rationale behind this working concentration ?

In my opinion, the topic of this paper is relevant, timely, and of interest to the audience of this journal. The paper is easy to follow, generally clear and free from grammatical or spelling errors. The manuscript is technically correct and the methods are used correctly so that it is in my opinion that the data is sufficient to corroborate the claims (as long as points a-b are addressed). The reporting appears to also be sufficiently transparent to repeat the experiments for those not familiar with these types of experiments. The graphs and figures with captions are appropriate and easy to comprehend and the supporting evidence and statistical data is reliable and properly validated. The direct role of DA to plasticity might require further studies in cell culture to unravel the molecular mechanisms involved. A next step in this series of investigation is to train rodents to vigorous physical activity and test the role for improved functional recovery with LD treatment after stroke with enhanced physical activity.

Author Response

Ms #ijms-1335292

Response to reviewer #2

Main Concerns

Point 1: “Olig2 shows phenotype overlap in the various developmental stages of the oligodendrocyte lineage and is featured from the oligodendrocyte progenoitor cell right up to the mature myelinating oligodendrocyte. Therefore, Olig2 is not entirely specific to adult oligodendrocyte precursors. The bipolar phenotype and more specific PDGF and A2b5 are more reliable markers that should have been used. Therefore it might be incorrect to state that in this study, LD treatment enhanced the number of oligodendrocyte precursor cells after MCAO.”

We agree with the reviewer. We have changed the text in this respect in lines 163 to 178, 194, 276 and 282.

Point 2: “It has been extensively shown from previous studies that pre-ols increase in number after MCAO and to the extent of even populating and spreading into the contralateral hemisphere depending on the extent of the severity of the insult. In that case, the authors must retract their notion that DA signaling is directly involved in oligodendrogenesis.”

We agree with the reviewer and we have modified the interpretation of our findings. Treatment with levodopa has a modulatory effect on oligodendrocyte precursor cells increasing the number of OLIG2 positive cells. We have changed the manuscript text accordingly.

Point 3: “The surprising fact that levels of GAP43 was found to be lower after tMCAO compared to sham and subsequently in the treatment group requires a more insightly perspective as this goes against the notion of sprouting and plasticity that are central in regeneration. It is is well established that GAP-43 levels of expression increase during regeneration in growth cones of axonal membranes. As a matter of fact, deletion of one allele of the GAP-43 gene in humans leads to cognitive impairment and failure to form telencephalic commissures.”

As we have mentioned in the Discussion (lines 264 to 275) the dynamics of GAP-43 expression is dependent on the time point GAP-43 is assessed following the insult. Differences in expression and levels might be obtained due to different detection methods i.e. immunohistochemistry/immunofluorescence, Western blotting or quantitative rtPCR, different stroke models in different species. Throughout the literature either a depletion of GAP-43 protein during the second and third week after experimental stroke (Gorup et al., 2015) was found while others found a transient increase of GAP-43 during the first weeks after stroke (Sist et al., 2014). At 30 days, three independent studies found protein levels of GAP similar or lower compared to levels in sham-operated animals (Sist et al., 2014; Zhao et al.; Li et al., 2019), others an increase i.e. (Gorup et al., 2015).

We agree with the reviewer in this points, however, further in depth studies will be required to evaluate the exact spatiotemporal expression of Gap-43 in different stroke models and species.

Minor concerns

Point 4: “It is unclear why the authors of this paper performed experiments on two different rodent stroke models that involved both mice and rats. Was it to show congruent results irrespective of animal spieces or was it to determine whether mechanistically their findings can be validated in stroke in general ? In my opinion this point has to be clarified for the sake of the readers of this journal.”

The rationale to include different models of stroke in different species was to provide the evidence of sprouting of dopaminergic fibers in the ischemic territory. We agree with the reviewer and pointed out this aspect in a way we hope it becomes more comprehensive for the reader.

Point 5: “Was the blood flow reduction in the MCA region and the lesion volumes comparable in all the experimental groups tested ? The authors should at least show some TTC-stained serial sections.”

 In all studies performed serial sections were stained for NeuN to measure infarct volume at the endpoint of the study. In none of the studies, differences in infarct volume were detected. Studies are referred to in the manuscript.

Point 6: “Section 2.3 should read: ‘Treatment with levodopa increases the number of adult oligodendrocyte precursor cells in the per-infarct area.”

Reviewer is correct, it has been changed.

Point 7: “LD was selected at a doseage of 20mg/Kg. What was the rationale behind this working concentration ?”

Dosages have been adapted to the clinical situation. In our initial studies, we started with 5 mg/kg, which represents a daily dosage of 350 mg in a person weighing 70 kg. In addition, we also performed dose-escalating studies up to 20 mg/kg per day (Ruscher et al., 2012). Treatment did not show any side or adverse effects. In addition, we also took in consideration previously published studies on pharmacodynamics and pharmacokinetics of the drug in rats (i.e. Bredberg et al., 1994).

References

Bredberg, E., Lennernäs, H., and Paalzow, L. (1994). Pharmacokinetics of levodopa and carbidopa in rats following different routes of  administration. Pharm. Res. 11, 549–555. doi:10.1023/a:1018970617104.

Gorup, D., Bohaček, I., Miličević, T., Pochet, R., Mitrečić, D., Križ, J., et al. (2015). Increased expression and colocalization of GAP43 and CASP3 after brain ischemic  lesion in mouse. Neurosci. Lett. 597, 176–182. doi:10.1016/j.neulet.2015.04.042.

Li, M.-Z., Zhan, Y., Yang, L., Feng, X.-F., Zou, H.-Y., Lei, J.-F., et al. (2019). MRI Evaluation of Axonal Remodeling After Combination Treatment With Xiaoshuan  Enteric-Coated Capsule and Enriched Environment in Rats After Ischemic Stroke. Front. Physiol. 10, 1528. doi:10.3389/fphys.2019.01528.

Ruscher, K., Kuric, E., and Wieloch, T. (2012). Levodopa treatment improves functional recovery after experimental stroke. Stroke 43, 507–513. doi:10.1161/STROKEAHA.111.638767.

Sist, B., Fouad, K., and Winship, I. R. (2014). Plasticity beyond peri-infarct cortex: spinal up regulation of structural plasticity, neurotrophins, and inflammatory cytokines during recovery from cortical stroke. Exp. Neurol. 252, 47–56. doi:10.1016/j.expneurol.2013.11.019.

Zhao, S., Zhao, M., Xiao, T., Jolkkonen, J., and Zhao, C. (2013). Constraint-induced movement therapy overcomes the intrinsic axonal growth-inhibitory  signals in stroke rats. Stroke 44, 1698–1705. doi:10.1161/STROKEAHA.111.000361.
